# A Deep Learning-Driven Self-Conscious Distributed Cyber-Physical System for Renewable Energy Communities

**DOI:** 10.3390/s23094549

**Published:** 2023-05-07

**Authors:** Giovanni Cicceri, Giuseppe Tricomi, Luca D’Agati, Francesco Longo, Giovanni Merlino, Antonio Puliafito

**Affiliations:** 1Department of Engineering (DI), University of Messina, 98122 Messina, Italy; gcicceri@unime.it (G.C.); luca.dagati@studenti.unime.it (L.D.); flongo@unime.it (F.L.); gmerlino@unime.it (G.M.); 2Department of Biomedicine, Neuroscience and Advanced Diagnostics (BiND), University of Palermo, 90127 Palermo, Italy; 3Department of Biomedical and Dental Sciences, Morphological and Functional Images (BIOMORF), University of Messina, 98122 Messina, Italy

**Keywords:** renewable energy communities (RECs), energy-aware DCPS, edge-to-cloud infrastructure, smart grids, Internet of Things, deep learning, energy management

## Abstract

The Internet of Things (IoT) is transforming various domains, including smart energy management, by enabling the integration of complex digital and physical components in distributed cyber-physical systems (DCPSs). The design of DCPSs has so far been focused on performance-related, non-functional requirements. However, with the growing power consumption and computation expenses, sustainability is becoming an important aspect to consider. This has led to the concept of energy-aware DCPSs, which integrate conventional non-functional requirements with additional attributes for sustainability, such as energy consumption. This research activity aimed to investigate and develop energy-aware architectural models and edge/cloud computing technologies to design next-generation, AI-enabled (and, specifically, deep-learning-enhanced), self-conscious IoT-extended DCPSs. Our key contributions include energy-aware edge-to-cloud architectural models and technologies, the orchestration of a (possibly federated) edge-to-cloud infrastructure, abstractions and unified models for distributed heterogeneous virtualized resources, innovative machine learning algorithms for the dynamic reallocation and reconfiguration of energy resources, and the management of energy communities. The proposed solution was validated through case studies on optimizing renewable energy communities (RECs), or energy-aware DCPSs, which are particularly challenging due to their unique requirements and constraints; in more detail, in this work, we aim to define the optimal implementation of an energy-aware DCPS. Moreover, smart grids play a crucial role in developing energy-aware DCPSs, providing a flexible and efficient power system integrating renewable energy sources, microgrids, and other distributed energy resources. The proposed energy-aware DCPSs contribute to the development of smart grids by providing a sustainable, self-consistent, and efficient way to manage energy distribution and consumption. The performance demonstrates our approach’s effectiveness for consumption and production (based on RMSE and MAE metrics). Our research supports the transition towards a more sustainable future, where communities adopting REC principles become key players in the energy landscape.

## 1. Introduction

The Internet of Things (IoT) has become increasingly prevalent across various application domains, such as smart cities and Industry 4.0, leading to a heightened emphasis on the design and development of distributed cyber-physical systems (DCPSs). These systems’ behavior is significantly influenced by their context, encompassing the external physical environment and the internal states of the IT components and networked infrastructure. In recent years, DCPSs have been proposed to facilitate renewable energy communities (RECs), which promote sustainable development within local communities by adopting renewable energy sources. RECs consist of individuals, organizations, and businesses collaborating to produce and consume renewable energy, such as solar or wind power. Integrating DCPSs in RECs can enhance energy usage efficiency by monitoring and controlling energy flow within the community. DCPSs provide the essential infrastructure for RECs to supervise and regulate the production and consumption of renewable energy sources. In this context, IoT devices collect energy production and consumption data, which is then analyzed by cloud-based platforms to optimize the energy management system. By harnessing these technologies, RECs can establish a more decentralized and democratized energy system, empowering local communities to manage their energy resources actively. Numerous global initiatives have successfully integrated DCPSs and RECs. For example, in Germany, the “EnergieWendeBauen” project (energiewendebauen.de, accessed on 1 March 2023) has implemented a DCPS-based platform for energy management in residential communities. This platform enables residents to monitor and control their energy usage and share excess renewable energy within the community. Similarly, the “Solar Share” project in Italy (lifegate.it, accessed on 1 March 2023)has introduced a DCPS-based platform that allows individuals and small businesses to share surplus solar energy with their neighbors. Integrating DCPSs and RECs offers a promising opportunity to promote sustainable development and transform the energy landscape. By leveraging the power of IoT and cloud computing technologies, these systems can enable more efficient, sustainable, and decentralized energy management. Future research in this area should focus on developing scalable and secure DCPS-based platforms to support the widespread adoption of renewable energy sources in local communities worldwide.

To enable the creation of DCPSs, an overlay-based distinction between the physical environment and the digital infrastructure is considered a cornerstone of the whole scenario. IoT devices’ sensing and actuation capabilities facilitate this interaction between the two layers, which collect data to send to the cloud for processing according to various scopes, such as latency reduction, privacy-preserving, or security purposes. Data processing in the cloud typically involves logic units adapting their models based on observed data and providing dynamic and queryable run time models for a pipeline of services. Until now, the design of DCPSs has primarily focused on performance-related, non-functional requirements. However, sustainability has become critical due to the growing power consumption and associated computing expenses at different levels in these systems. The increasing sophistication of DCPSs requires more computational resources, which leads to increased energy costs. To address the sustainability challenge, integrating energy-aware digital components in DCPSs is an essential activity to create sustainable systems where IoT devices and server-based infrastructures can make autonomous decisions based on the outcomes of self-learning algorithms. DCPSs are becoming increasingly complex and consist of multiple interacting subsystems and environments. The aggregation of subsystems occurs at different levels, from edge devices to large systems. The proposed solution envisions a future where DCPSs are treated as conscious systems that can respond to internal and external triggers and adapt their operations to achieve predefined goals. These systems will be able to learn from experience through self-learning mechanisms and carry out planned actions and predictive strategies at the overall system level to optimize resources, maximize efficiency, and reduce energy costs.

A renewable energy community realized upon a DCPS is an environment in which two aspects must be combined and orchestrated: energy production and energy consumption. The trade-off has to be realized not only in terms of online orchestration but also by considering historical data related to the two aspects mentioned above. From this perspective, IoT devices are essential to observe physical parameters, such as current consumption and voltage. A distributed infrastructure collects and processes these samples through optimizing self-learning algorithms.

The goal of the envisioned environment is to ensure the optimal behavior of the entire REC by maximizing self-consumed energy and minimizing the delta between the community’s produced and its consumed energy profiles. This way, the presented solution is tailored to a scenario in which the renewable energy community is composed of real estate units agreeing to create a DCPS in which they cooperate with each other through a central entity appointed to act as an energy manager and broker toward the grid. The broker, running on the DCPS server facilities, has three main duties:Distributing the energy produced among the whole community, avoiding purchasing energy from the grid as much as possible;Continuously monitoring the energy market to purchase and sell the energy at the best price;Notifying the REC’s end users, suggesting disconnecting specific submetering or lightening the energy load to comply with the consumption parameters defined by the REC to obtain a better monetary reward. The reward (obtained by the REC concerning the energy available for sale) is shared proportionally to the correct user behavior. The proposed methodology incentivizes efficient energy use and contributes to a more sustainable energy ecosystem.

Of course, these aspects and considerations are not the only elements relevant to cope with this goal. Indeed, REC designers also have to consider other factors, such as:The placement of the computation entities inside the infrastructure;IoT and infrastructure management;Environmental energy predictions: production and consumption;User data privacy.

In this work, we present a self-conscious system designed to constantly monitor and forecast energy consumption in real estate units under the purview of the REC. This groundbreaking approach facilitates energy management and reduces dependency on external power grids. To highlight the contributions:(i)Our system uses advanced deep learning algorithms to accurately predict energy production and consumption patterns, paving the way for more efficient and eco-friendly energy distribution.(ii)When energy consumption exceeds production, the system proactively dispatches notifications to real estate units, indicating high consumption levels to those estates with elevated projected consumption. This timely communication encourages residents to shorten their energy usage, ultimately reducing the necessity of procuring supplementary energy from the grid.(iii)The system meticulously records and examines the responses to these reduction requests to support community involvement and commitment. These data are later shared with the community, which can then deliberate a reward-based incentive program to recognize those who consistently exhibit responsible energy consumption practices.

This approach favors energy efficiency and sustainable living within the REC by fostering a cooperative atmosphere, providing mutual benefits for all community members.

This paper is organized as follows. In Section 2, we provide a comprehensive literature review of the existing methods and technologies used in energy management and discuss the advantages and limitations of each. Section 3 provides the background information necessary to understand the presented work. In Section 4, we describe the system architecture and the role of each component in detail. Section 5 presents a case study illustrating the practical application of the proposed solution, while Section 6 discusses the results and validation of our approach. Finally, in Section 7, we conclude with our findings and suggest future enhancements to improve the effectiveness and applicability of our solution.

## 2. Related Works

Distributed cyber-physical systems (DCPSs) can significantly benefit from recent advancements in distributed computing, including architectural elements, algorithms, and models. In [1], the authors highlight key challenges associated with DCPSs, such as latency, energy consumption, security/privacy, and reliability. Designing a reliable IoT communication infrastructure for DCPSs remains an open challenge, as other researchers in [2,3] emphasized. Meanwhile, ref. [4] formulates the scheduling computation on the cloud continuum as a mixed-integer linear programming problem and proposes an energy-aware deployment and replication scheduling model, considering the capability of edge/fog nodes to harvest “green” energy.

The increased adoption of DCPSs, combined with the need to address emerging climate change issues, has led to renewable energy communities (RECs). In recent years, energy delivery and consumption in DCPSs have gained particular attention due to the increasing number of users (producers and consumers) involved in generating and sharing renewable energy [5,6]. Research on energy management and optimization through energy exchange, sharing, and storage mechanisms, along with the characterization of user behaviors, is crucial for achieving sustainability in RECs [7,8,9]. In this context, ref. [10] proposes a distributed energy management system (EMS) for optimal microgrid operation, considering power distribution constraints. The EMS demonstrates effectiveness in both islanded and grid-connected modes, with future work focusing on its implementation in real systems and performance analysis. Cloud computing has emerged as a popular solution for managing, storing, and processing data in energy systems. As outlined by [11], it offers a scalable, on-demand, and cost-effective model for delivering IT resources via the Internet. Numerous researchers have investigated the application of cloud computing for energy management and optimization. In [12], the authors explore the new challenges that smart grid technology introduces for comprehensive data management and examine how cloud computing can address these issues. Their survey encompasses smart grid and energy management methods, investigating the use of cloud computing in various domains, such as energy management, demand-side management, building energy management, energy hubs, and power dispatching systems.

Smart grids represent a modernized electrical grid infrastructure that employs cutting-edge technologies to monitor, control, and optimize electrical power generation, distribution, and consumption. The authors in [13] present a detailed overview of smart grid technologies, including advanced metering infrastructure, demand response, and distributed energy resources. Furthermore, ref. [14] reviews demand-side management techniques in smart grids, emphasizing the importance of load forecasting, demand response, and energy storage systems in achieving energy efficiency and grid reliability. Integrating the Internet of Things (IoT) and cloud computing has shown immense potential in enhancing the efficiency of energy management systems. IoT provides a platform for connecting and collecting data from various devices and sensors, while cloud computing enables the processing and analysis of these data. In [15], the authors discuss how incorporating IoT technologies into smart grids can improve monitoring, communication, and data processing across various devices. They propose a layered approach for classifying IoT applications in smart grids and explore recent research efforts along with future directions. On the other hand, the authors in [16] investigate the benefits of combining IoT and cloud computing for smart grid applications, particularly in demand response, fault detection, and renewable energy integration. This synergistic approach holds promise for further energy management and optimization advancements, paving the way for more sustainable and efficient energy systems. Ref. [17] investigates the correlation between solar irradiance and harmonic distortion in grid-tied photovoltaic distributed energy resource (PV-DERs) systems. Understanding this relationship can help develop effective grid-to-grid power-sharing arrangements and mitigate harmonics in bidirectional power-transfer community-grid structures.

The self-management processes that govern the operation of RECs are based on machine learning (ML) techniques to improve their effectiveness, autonomy, and efficiency. Energy demand and supply forecasting, self-consumption, characterization of power consumption behaviors, efficient scheduling of energy resources, and appliance obsolescence are some tasks involving ML and deep learning (DL) techniques [18,19,20,21].

Some studies have been conducted using both statistical approaches [22,23,24] and ML models for predicting individual household loads, predominantly the latter, due to their ability to capture complex patterns in the data and provide accurate predictions [25,26,27,28]. On the other hand, despite other works that have been conducted to improve the accuracy of household load forecasting using the advantages of DL models, and thus of the use of the neural network (NN)-based algorithms [29,30,31], other investigations have focused on improving the accuracy of household load forecasting by taking advantage of DL architectures for time series prediction, including the highly effective long short-term memory neural networks (LSTMs) [32,33]. The latter have demonstrated remarkable advancements in recent times, despite the volatility of predictions caused by the heterogeneity and randomness of household behavior; however, they are out-performed by the more accurate Bi-LSTM networks [34,35,36]. In this context, modeling user profiles to meet energy demand while optimizing overall consumption is crucial [37]. Thus, DL models are a must to identify users’ lifestyles based on their daily energy consumption. In addition, the meteorological forecast data must also be considered when modeling energy profiles, as renewable energy sources are often intermittent. Research on developing planning strategies for smart load distribution and integrating renewable energy resources is ongoing, and federated learning (FL) approaches are being investigated for this purpose [38,39].

Energy awareness must be incorporated at every layer (models, data, algorithms, hardware components, etc.) and tier (cloud, edge/fog, IoT) of the IT infrastructure of DCPSs, and in every phase (design, deployment, execution, etc.). To address this problem, the scientific community has begun to define methodologies and approaches to evaluate the energy consumption of models and algorithms based on structural and behavioral parameters [40]. For example, ref. [41] proposes an energy-efficient IoT data compression algorithm to optimize the execution of ML algorithms at the edge. At the same time, ref. [42,43] focuses on the energy optimization of the deployment and distributed training of ML models at the edge, respectively. The processing capabilities of IoT devices represent both a resource and a constraint. Thus, designing a suitable infrastructure is both a requirement and a challenge. The trend towards offloading data analytics tasks from edge devices to the cloud has been increasing. However, existing offloading approaches face the challenge of being static and needing help to adjust to changing workloads and network conditions.

Moreover, in [44], an energy-aware workload allocation framework for distributed deep neural networks (DNNs) in the edge-cloud continuum was presented to minimize energy cost for inference. This framework considers energy consumption and computation performance to optimize the allocation of workloads in a distributed computing environment. Offloading data analytics tasks from edge devices to the cloud has great potential for improving the efficiency and performance of DCPSs. However, existing offloading approaches have limitations, and researchers continue to develop more dynamic and energy-efficient solutions to overcome these challenges.

The advancements in DCPS research make significant progress on latency, energy consumption, security/privacy, reliability, and computation allocation challenges, improving their effectiveness, autonomy, and efficiency while contributing to sustainability and addressing emerging problems related to climate change. For these reasons, the solution proposed in this study aims to define an optimal implementation/architecture of an energy-aware DCPS, providing a smart and flexible power system while enabling the integration of renewable energy sources and facilitating the integration of microgrids and other distributed energy resources. Ref. [45] presents an asymmetrical single-phase eleven-level inverter for the grid integration of distributed power generation sources, contributing to improved power quality and cost effectiveness in grid-connected systems. Moreover, ref. [46] introduces a distributed-variable flow-variable temperature (VF-VT) approach for integrated energy and heating systems, offering privacy preservation, feasibility, and scalability. The study identifies future research directions, including global optimization, model development, and improved thermal dynamics modeling, which can further enhance the performance and efficiency of energy-aware DCPSs.

Our proposed solution employs a combined approach for managing both the production and consumption aspects of RECs, which sets it apart from other systems. In addition to this comprehensive approach, our solution provides three key contributions that, although present in some existing solutions, are not typically found together in a single framework. Specifically, our approach integrates all three contributions, enhancing the overall effectiveness and efficiency of the system. In comparison, the papers from references [18,19,20,21,22,23,24,25,26,27,28,29,30,31,32,33,34,35,36,37] primarily focus on applying AI techniques to individual households rather than entire communities. While these studies offer valuable insights into AI-based energy management, they may not fully capture the complexity and interconnectednessof energy production and consumption in broader communities. By addressing energy management at the community level, our solution aims to achieve a more comprehensive understanding and optimization of energy distribution and utilization in RECs. Moreover, the works from references [5,6,7,8,9] do not explicitly mention the use of AI techniques in their proposed solutions. Although these studies contribute to advancing energy-aware DCPSs, they may not fully leverage the potential of AI and ML in improving energy management, forecasting, and optimization in RECs. By incorporating AI and, more specifically, DL techniques in our solution, we seek to further enhance the performance, efficiency, and adaptability of our proposed energy-aware DCPS architecture.

## 3. Background

### 3.1. 
Stack4Things: Integrating IoT
Resources
into OpenStack
as I/Ocloud


Stack4Things (S4T) [47] is an open source research project and innovative platform designed to extend the widely used cloud management system, OpenStack, into the Internet of Things (IoT) realm. S4T aims to facilitate the management of IoT and edge device deployments within the OpenStack ecosystem, implementing appropriate features to seamlessly integrate IoT infrastructures into the edge-extended Infrastructure-as-a-Service (IaaS) and Platform-as-a-Service (PaaS) clouds. Furthermore, the Input/Output (I/O) cloud [48] approach leverages S4T functionalities to provide standardized and generic programming capabilities on top of IoT resources, independently of the underlying infrastructure configurations.

#### 3.1.1. S4T Architecture and IoT Management

The S4T architecture primarily consists of a cloud-side component, IoTronic, and one or more edge-side components called Lightning-Rod (LR). These components enable users to utilize IoT devices and their I/O resources, such as sensors and actuators, through well-defined APIs similar to those available for standard cloud resources. This I/O cloud concept offers IoT virtualization features alongside traditional IaaS (computing and storage) virtualization. On the other hand, virtual nodes (VNs) host the business logic and use the attached I/O resources, emulating real IoT devices. S4T’s IoT management involves various OpenStack subsystems, with IoTronic as a central component responsible for provisioning and configuring IoT nodes with embedded sensing and actuation resources. Neutron’s OpenStack networking service has been enhanced to ensure seamless connectivity for IoT nodes deployed at the network edge. Additionally, the platform leverages the integration of Zun and Qinling to enable Function-as-a-Service (FaaS) capabilities. Zun provides container management, while Qinling is the FaaS subsystem, streamlining container deployment and orchestration. Together, these subsystems create a comprehensive and efficient IoT management solution.

#### 3.1.2. IoTronic Cloud-Side Service

The IoTronic cloud-side service is a crucial component of the S4T architecture, designed with modularity, scalability, and robustness. As illustrated in Figure 1, IoTronic’s primary function is to manage and orchestrate seamless connectivity between edge devices and the cloud, providing users with a comprehensive interface for managing IoT devices remotely. It extends the OpenStack architecture toward managing sensing and actuation resources, aligning with the Sensing-and-Actuation-as-a-Service (SAaaS) paradigm. IoTronic interacts with the LR device-side agent to establish and maintain a reliable connection between the cloud and the edge devices, even in the presence of network address translations (NATs) or strict firewalls. This connection is facilitated through WebSocket technology, which employs the Web Application Messaging Protocol (WAMP) to create a full-duplex messaging channel.

The cloud-side architecture comprises several components, including the IoTronic Conductor, which manages the IoTronic database that stores essential information, such as unique device identifiers, user and tenant associations, device properties, and hardware/software characteristics. The IoTronic APIs expose a REST interface for end users, allowing interaction with the service via a custom client or a web browser. The OpenStack Horizon dashboard has been extended with a Stack4Things dashboard, offering access to all functionalities provided by the IoTronic service and other software components. IoTronic also features a WAMP agent that bridges other components and edge devices, translating Advanced Message Queuing Protocol (AMQP) messages into WAMP messages and vice versa. This design makes the architecture highly scalable, as components can be deployed on different machines without impacting service functionalities. Additionally, it ensures redundancy and high availability for IoT systems by allowing multiple IoTronic WAMP agents and WebSocket tunnel agents to be instantiated, each managing a subset of IoT devices.

#### 3.1.3. Lightning-Rod Device-Side Agent

The LR device-side agent is an essential component of the S4T architecture, characterized by its modularity, fault tolerance, and streamlined design. The architectural structure of the LR agent is depicted in Figure 2. Its primary function is to facilitate seamless connectivity between edge devices and the S4T IoTronic service, even when deployed behind NATs or under the constraints of stringent firewalls. This connectivity is achieved through WebSocket technology, which employs the WAMP to establish a reliable, full-duplex messaging channel between the cloud and the devices.

#### 3.1.4. S4T Features

S4T offers support for a variety of features, including:*Authorization and Authentication*: S4T manages user authentication using OpenStack’s identity service, Keystone, and grants authorization for accessing and controlling remote IoT devices.*Remote Access and Management*: Users can access their IoT devices without concern about location or networking configurations, thanks to S4T’s WebSocket-based reverse-tunneling mechanism.*Remote Customization and Contextualization*: S4T allows users to define the application logic to be executed on devices, which is then distributed as functions and deployed on IoT devices under authorization and privacy policies, even during runtime. Finally, S4T supports Python and Node.js runtime environments.

#### 3.1.5. I/O Cloud: Seamless Integration of IoT and Cloud Resources

As mentioned, the I/O cloud approach, leveraging S4T functionalities, aims to provide standardized and generic programming capabilities on top of IoT resources, independently of the underlying infrastructure configurations. This approach maintains the ability to employ the unique characteristics of an IoT-enhanced distributed data center, such as the availability of edge nodes, which can now be used as computing infrastructure for data (pre)processing. Consequently, I/O cloud aims to achieve seamless integration between the cloud and IoT by offering distributed IoT resources (i.e., sensors and actuators) hosted on edge nodes as virtualized cloud resources. A crucial aspect of the I/O cloud approach is ensuring that IoT deployments function as active components within the cloud infrastructure while maintaining their unique characteristics. It must provide efficient I/O virtualization (virtIO) to achieve this. By extending the concept of virtualization to the IoT domain, I/O cloud abstracts IoT resources, presenting them as virtual entities. These virtual resources can be accessed through a user-friendly interface that reflects the I/O primitives of their physical counterparts. This highly customizable abstraction process allows users to encompass the entire I/O resources of an IoT node or only a specific subset of these resources. Additionally, it enables the logical consolidation of IoT resources from various nodes within a single (logical) entity. This flexibility allows developers to manage and integrate IoT resources within the cloud infrastructure efficiently, simplifying the process and enhancing the overall system’s usability. I/O Virtualization is based on file system virtualization to provide a virtual representation of pins of a physical IoT node while hosting user-defined logic and facilitating interactions with remote physical IoT resources simultaneously. Technically, an I/O cloud instance is a self-contained, isolated environment with a user-space-defined file system, sysfs. This is realized using FUSE technology over remote procedure calls (RPCs) to ensure remote interactions with the physical IoT resources.

### 3.2. Environment 4.0: Smart and Self-Conscious Environment by Design

On the way to a self-conscious environment by design, the research conducted by our group is exploring and paving the way for the realization of a self-managed environment, with a high degree of decoupling thanks to adherence to software-defined principles. This approach, named *Software-Defined CPS Function Virtualization* (SDCPS-FV), introduced in [49] as an underneath approach defining a software-defined city infrastructure, aims to distribute the CPS operations (such as control procedures, actions, and reactions to environmental inputs) along the whole CPS infrastructure. The SDCPS-FV is an enabler for the definition of CPSs, self-(re)configurable due to its ability to deploy IoT/Edge node control logic to specific devices in the CPS. These principles match perfectly with the definition of distributed cyber-physical systems described in Section 2, representing a suitable bedrock for setting up a self-conscious environment. An example of this is presented in [50], where the CPS autonomously re-configured the logical device connection to preserve the fire protection system functionalities (an example of a self-conscious environment realized on SDCPS is shown in Figure 3).

Nevertheless, in the previous example, the *self-awareness* of the environment is presented in terms of system resiliency obtained by hierarchical management controls distributed on the CPS that spans from the data-center/cloud systems to the edge devices through the fog devices by intermediation, and the resulting CPS monitors its behavior and reacts to the (negative) events identified. The ability to react to events or situations is a key point for self-conscious environments. In this context, an event could be a predefined situation, such as people moving from one building to another as in [51,52], or unexpected issues that a system has to face (as described in [50]); the events are perceived and managed by the environment autonomously, which hosts specific algorithms, specialized controllers, or AI modules.

## 4. Architecture

The REC envisioned is a DCPS in which it is possible to consider an architecture composed of three parts by design: (i) the REC production sites, (ii) the REC real estate units, and (iii) the centralized computation facilities.

The “REC production sites” may be realized in several ways, spanning from a dedicated community’s solar panel field to a distributed production system in which the panels are installed on the properties belonging to the REC. Indeed, the ownership of energy production systems is not a key point in the scenario; this is because the only essential element is that all the sources are connected together and are able to deliver some operative information enabling the DL model to predict and monitor the energy production.

The “REC real estate units”, considered one by one, are no more than a sub-CPS composed of several IoT devices (smart meters or powerful devices with metering capability). This way, a building in this category is continuously monitored by a DL model running on site (essential to preserve the owner’s privacy, e.g., the owner’s habits), producing aggregated data to the central coordinator (referring to the whole building). The data monitored in a real estate unit may refer to a section or simply to a particular device (i.e., the washing machine). Furthermore, the data gathered are stored locally and processed by the DL model to predict consumption, considering the seasonality.

The centralized computing facilities can run the DCPS orchestration modules managing the whole REC. The data coming from the DCPS to the orchestrator arrive in the form of real-time and predicted data; these contributions are processed to compute the amount of energy available to the community at the computation time and in the following time window. This way, the orchestrator can evaluate the community energy needs and, if needed, notifies to REC members that have, both real-time and predicted, higher consumption, to reduce the energy request in their real estate units. The acceptance or lack thereof of community members receiving a notification is stored for administrative duties (e.g., to reward deserving members with discounts or benefits). Centralized computing facilities also have computed DL-based monitoring and prediction estimation related to “REC production sites”.

The REC’s energy orchestrator, as shown in Figure 4, is composed of three main modules. The first two (respectively, named ECEM, Energy Consumption Estimation Module, and EPREM, Energy Production Estimation Module) are meant to evaluate both real-time data and predicted data to produce an approximated value of consumption (or production); the third (called TEANS, Threshold Evaluator and Notification System) uses the previous approximation to understand if the energy requests are greater than available energy in the community, and tries to find a solution pattern to avoid it (i.e., in the case of unexpected consumption, one or more notifications may be delivered to the community members with higher consumption).

### 4.1. REC Subsystems at the Edge: Real Estate Units and Production Sites

Both the real estate units elements and the production sites, as stated above, may be considered as sub-DCPSs composed of a series of IoT devices configured and managed by the exploitation of the principles (and the IoT platform S4T) described in Section 3 and Section 3.1, which enable the system owner (and, of course, the manager) to inject portions of code in the form of plugins to customize the IoT’s behavior. Furthermore, thanks to the software-defined approach (as described in [49]), it is possible to orchestrate the environments as configurable DCPS contained inside the community’s DCPS. The two edge elements of the REC are monitored (in terms of energy consumption and energy production) by a DL model trained on the historical data collected on the system itself (indeed, the model is trained on the historical data year by year to increase its accuracy in prediction, also in relation to the environmental modification such as the increment or replacing of devices). The monitoring DL model can predict the system behavior (energy consumption or production) by advancing a time slot Δt. According to Figure 4, the real estate units and the production sites present a different management approach: the former runs its monitoring model in each estate unit, providing data (real-time and prediction) to the opportune orchestrator sub-module (ECEM). Conversely, the latter runs only one model on cloud computing facilities that can process and gather the data coming from each production site and predict the energy produced. Data obtained from the aggregation and prediction duties are sent to the orchestrator sub-module (EPREM). Furthermore, this approach avoids privacy disclosure by processing the data about energy consumption directly on the real estate units and pushing only the aggregated data towards the REC energy orchestrator. From the security point of view, the architecture also increases the security degree by decoupling the home energy management system and REC energy management system.

### 4.2. The REC Energy Orchestrator

The energy orchestrator is a system meant to compare energy consumption and production, aiming to avoid the necessity to buy energy from the grid. To reach this goal, the contributions related to real-time data and predicted data are computed by specific sub-modules (e.g., ECEM and EPREM) to produce an estimated value of both energy requests and energy available for the community. The EPREM (Energy Production Estimation Module) and the ECEM (Energy Consumption Estimation Module) deliver their estimation to the TEANS (Threshold Evaluator and Notification System) which, if needed, sends notifications to the members of the community with higher consumption.

The estimations are based on an assumption of linear behavior among data from the monitoring and prediction, both for consumption and production. This assumption is supported in [53], where a linear model based on electricity consumption data was sufficient to forecast industrial production. The investigation mainly focused on forecasting Italian industrial production, where results implied that linear behavior is a valid assumption for short-term forecasting, 50 min forecasts in this case (see Section 5.1 related to the datasets). Based on the previous assumption, the estimation modules use Equation (Equation 1) to compute the $x_{\left(est\right)}$ as a point on a straight line passing through two points, real-time data ($t_{0}$) and predicted data ($t_{\Delta }$), computed at time $t_{est}$, as reported in Figure 5. The $t_{est}$ is configurable by the orchestrator value narrow to the real-time data.
(1)$$x_{\left(est\right)}=x_{\left(t_{0}\right)}+\frac{\left(x_{\left(t_{0}\right)}-x_{t_{\Delta }}\right)}{\left(t_{0}-t_{\Delta }\right)}*t_{est}-\frac{\left(x_{\left(t_{0}\right)}-x_{t_{\Delta }}\right)}{\left(t_{0}-t_{\Delta }\right)}$$

The two modules, ECEM and EPREM, have a dual behavior because the estimation process is used only in cases with positive (see Equation (Equation 2)) and negative (see Equation (Equation 3)) slopes. This way, the estimated values may not underestimate the consumption or overestimate the production.

The job performed by ECEM and EPREM is essential to enable the orchestration process so that we can consider their job as a pre-processing of the inputs from the edge part of the DPCS.
(2)$$x_{\left(estC\right)}=\left\{\begin{array}{cc} x_{\left(est\right)} & :(x_{\left(t_{0}\right)}-x_{t_{\Delta }})\geq 0\\ x_{\left(t_{0}\right)} & :(x_{\left(t_{0}\right)}-x_{t_{\Delta }})<0 \end{array}\right.$$
(3)$$x_{\left(estP\right)}=\left\{\begin{array}{cc} x_{\left(est\right)} & :(x_{\left(t_{0}\right)}-x_{t_{\Delta }})\leq 0\\ x_{\left(t_{0}\right)} & :(x_{\left(t_{0}\right)}-x_{t_{\Delta }})>0 \end{array}\right.$$

The estimated values become inputs of the TEANS (Threshold Evaluator and Notification System) module which, after evaluation of the inputs received, is able to:Identify when consumption is exceeding production;Identify which community members exhibit higher consumption; andSend notifications to the owner of a real estate unit and, at the same time, to the environment itself, that suggest which internal line is requesting more energy.

The function shown in Algorithm 1 is used in the first and third tasks listed above. Indeed, the function *EvaluateInput* is used by TEANS firstly to evaluate the inputs coming from ECEM and EPREM to understand if an intervention is necessary (the intervention procedure is contained in the else condition). When a reduction in energy requests is identified, the energy consumption that exceeds the energy available in the REC is computed (the δ value), and it is used to select the REC members who will be the recipients of notifications of power reduction. The quantity of requested power reduction notifications is split equally among the selected members, corresponding to the γ value. The second function to consider is shown in Algorithm 2. It is invoked by Algorithm 1 to obtain a list of members requesting high energy. To this aim, the input of this function is the δ mentioned above; the ECEM module uses it to understand how many members to select; the higher the value of δ, the lower the threshold used to mark and add a member to the list becomes. In any case, the ECEM also identifies a list of unusual consumers who generally do not request much power from the REC. In that case, they are excluded from the “powerAbsorbingConsumer” list. Furthermore, the REC energy orchestrator considers each time a reduction in energy consumption request is satisfied, which may be used for rewarding purposes in administrative processes.
**Algorithm 1:** Pseudocodeof the function used to evaluate inputs from estimation modules aiming to identify the necessity of reducing the REC’s energy requests. 
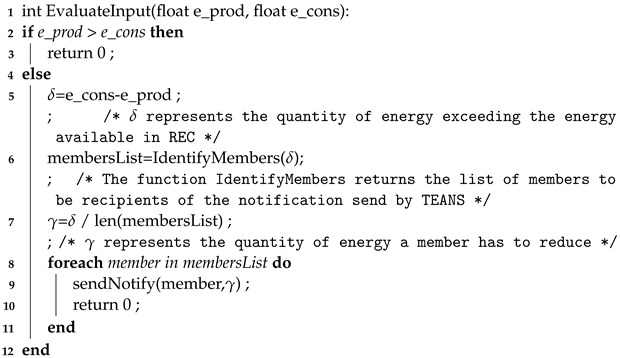

**Algorithm 2:** Pseudocode of the function used to identify the list of REC members that will receive the request for power reduction. 
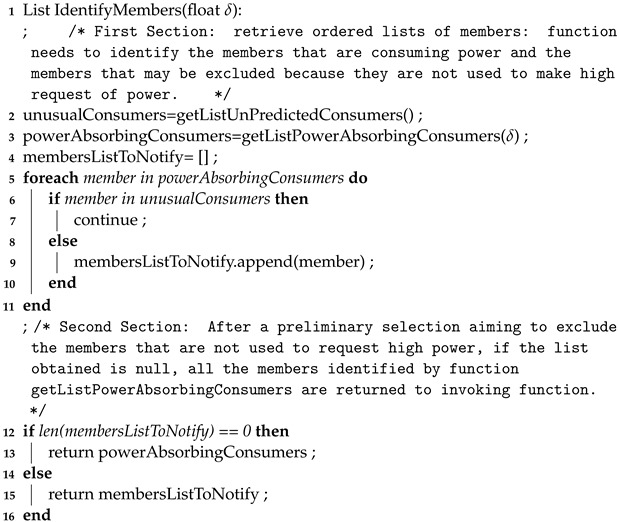



## 5. Use Case/Reference Scenario

This section describes the use case and a detailed analysis of the dataset, followed by an explanation of the preprocessing methods employed and, finally, an overview of the DL models used in the implementation of the system.

The use case and experiments conducted to analyze the behavior and feasibility of a REC implemented as described above are presented in the following. The REC community consists of numerous real estate units, enough to justify the production line realized in the REC. Each unit is equipped with a general smart meter, and the electrical systems within these buildings are intentionally designed to be divided into multiple subsections. These subsections are connected to specific sub-meters, allowing for more granular monitoring and management of energy consumption. The smart meters operate alongside IoT devices, such as Raspberry Pi 3, Arancino, and other CPU- and MCU-based edge devices, which are powerful enough to manage the edge component of S4T (Lightning-Rod, see Section 3.1.3). The DL models are deployed as plugins injected via S4T on these edge devices. Another aspect of the REC community involves the production sites, which consist of solar panels connected to an inverter to establish a production line. Generally, each production line may be located on a real estate unit (e.g., on a building’s roof) or in a designated area within the REC reserved for energy production. In this use case, we consider a few powerful production lines set up in specific areas of the REC. Each line has its dedicated inverter, which measures the produced DC and other parameters, as discussed in more detail in Section 5.1 and Section 6. These measurements are then transmitted to the DL model running in centralized computing facilities directly managed by the community. This setup allows for efficient monitoring and management of energy production within the REC community. A comprehensive understanding of the entire architecture, including the interconnections and interactions among all its components, can be obtained by referring to Figure 6. This chart offers a high-level overview of the relationships between the various elements within the system.

### 5.1. Dataset Description

For the use case, we used two different public datasets: (i) The household electricity load diagrams 2011–2014 dataset, designed by the University of California, School of Information and Computer Science, and shared in the UCI ML repository [54], and (ii) The solar power generation dataset [55]. The first dataset comprises 2,075,259 instances of *electricity consumption* samples (KW) of a household in Paris from December 2006 to November 2010 that were captured at 1 min intervals. It contains observations on household *global active power* (GAP) (in kilowatt), *global reactive power* (in kilowatt), *voltage* (in volts), *global current intensity* (in ampere), and information on three sub-rooms (in kilowatt/h) on global energy consumption (in kilowatt/h). Specifically, *Sub metering 1* monitors the active electricity usage of the kitchen appliances, including the dishwasher, oven, and microwave. *Sub metering 2* measures the active energy consumption of laundry room appliances such as washing machines, tumble dryers, and lights. *Sub metering 3* records the active power of the electric water heater and air conditioner.

The second dataset describes data gathered from two *solar power plants* in India over a 34-day period and includes power generation data and weather sensor readings data gathered every 15 min. For power generation plants, it provides readings on the amount of *DC power* (in kilowatt), the amount of *AC power* (kilowatt), and the *daily yield* and *total yield* for the inverter until that point. The weather sensor plants provide data on the *plant*, the *sensor panel id*, the *ambient temperature*, the *temperature* reading for that solar panel module, and finally, the *irradiation*. The power generation data are collected at the inverter level, where each inverter is connected to multiple lines of solar panels, while the weather sensor data are collected at the plant level, with a single array of sensors optimally placed at the plant.

Figure 7 highlights the seasonality of the GAP feature, used as the output value for the BiLSTM model’s predictions. Upon examining the data, it becomes evident that higher GAP values are concentrated in particular periods. The plot underscores the cyclical nature of the consumption patterns, facilitating the identification of intervals marked by increased energy usage. Data points in the graph display a color gradient from light to dark, representing a 24 h time frame arranged by the color intensity and organized into hourly, monthly, and weekly segments throughout the year. The hourly analysis indicates that consumption peaks predominantly occur during the afternoon, specifically from 17:00 to midnight. Conversely, the monthly overview offers insights into the distribution of consumption hours across the year, revealing a significant decrease in energy use during the summer months due to longer daylight hours. Additionally, a strong correlation between the monthly and weekly charts can be observed, as both show reduced average consumption in their central regions, corresponding to the summer period. This seasonality chart allows for the prediction of consumption patterns across various time frames during the year, assisting in the optimal configuration of the proposed system, aiming to predict the appropriate periods or hours to send notifications to users requesting a reduction in energy consumption. This approach aligns with RECs’ energy efficiency and sustainability policies.

Generally, a REC community consists of numerous real estate units, and they are commonly supported by energy generation sites to support the REC consumption. According to the data from the two datasets used for the experiments, we defined a reasonable scenario with 250 real estate units (as an assumption, we considered buildings with similar behavior and structure). This way, the energy produced by the three production lines from the datasets [55] used as our REC production site data is justified.

#### Data Preprocessing

Data preprocessing involved preparing the raw data for our analysis and using it as input for the recurrent models. This process consisted of several tasks, including data interpolation, reduction, normalization, and integration.

*Data interpolation*: We performed linear interpolation on the data by estimating missing or NaN (Not a Number) values in the *consumption* dataset and computing the value at each missing point as a linear combination of its neighbors;*Data reduction*: We aggregated time-series data into more manageable intervals by using the mean of every 15 min interval. This method facilitated the visualization and analysis of the patterns in the data and association with the *production* dataset;*Data normalization*: We normalized the data so that they fell within a range (0,1), so as to improve the performance of the DL models;*Data integration*: We aggregated the data from the *production* dataset by integrating the power generation and sensor readings data by feeding them as multivariate sequence inputs to the LSTM model.

Finally, we prepared entire time series data for use in a supervised learning approach by scaling the data and transforming them into a format suitable for training and testing the recurrent models. We divided the data into training, validation, and testing datasets to train the model, optimize hyperparameters, prevent overfitting, and test the final models’ performance. We split both datasets to use 70% of the data for training, 10% for validation, and 20% for testing.

### 5.2. Deep Learning Models

In our use case, we developed two types of DL models, and we tested them on the two datasets described above: long short-term memory (LSTM) and bidirectional long short-term memory (BiLSTM) neural network-based models, useful for processing sequential data and capable of learning long-term dependencies. Specifically, LSTM networks are specialized recurrent neural networks (RNNs) initially designed to overcome the vanishing gradient problem [56]. LSTM networks can preserve past information in sequential data, which results in precise forecasting for time-series datasets [57]. The BiLSTM networks, an extension of the LSTM networks, were applied twice to the input data to improve the long-term dependency learning and model accuracy [58]. The first LSTM was applied to the input data in their original order, while the second LSTM was applied to the input data in reverse order. In this way, the model can better retain information from both the past and future of the input sequence, resulting in improved accuracy. The main task of these networks is to learn potential rules from many samples of time-series data and perform analysis and predictions by constantly correcting the network weights. Both networks have significantly improved in recent years, especially for time series prediction and in analyzing power grid data [59,60], where the predictions perform well with both data generated from routine user behavior that exhibits some regularity and data generated from emergent user behavior, such as sudden incidents and anomalies, that show some randomness and variability.

The BiLSTM with an “attention mechanism” (BiLSTM-Attention) is particularly well suited for the analysis of long time series, due to a memory function that allows important feature information to be retained for load prediction [61,62]. The “attention mechanism” is used to further explore the relationship between the features of the predicted time points processed by the BiLSTM layer. This way, BiLSTM dynamically weighs the importance of different parts of an input sequence when making predictions, essentially by using a separate neural network to compute a set of weights representing each element’s importance in the input sequence. This is particularly useful in a household electricity consumption prediction at different times, as in cases where some parts of the input sequence are more important than others for making accurate predictions, e.g., in our use case, where the consumption during weekdays is lower than holiday periods, or consumption during spring is lower than summer, and so on. Thus, studying the consumption patterns at different time points can improve the accuracy of predictions. In the presented context, the BiLSTM-Attention networks were used as the proof of concept for the electricity consumption prediction for a REC community. In contrast, we used a simple LSTM network for plant generation data prediction (production), where it was assumed that the production data had constant behavior over time.

To evaluate the effectiveness of our models, we employed classical evaluation metrics, the root mean square error (RMSE), and the mean absolute error (MAE), where lower values represent better forecasting results. The equations for these metrics are given by Formulae (4) and (5), respectively.
(4)$$RMSE=\sqrt{\frac{1}{n}\sum_{i=1}^{n}{\left(y_{i}-\hat{y_{i}}\right)}^{2}}$$
(5)$$MAE=\frac{1}{n}\sum_{i=1}^{n}\left|y_{i}-\hat{y_{i}}\right|$$
where $y_{i}$ is the true value, $\hat{y_{i}}$ represents the prediction value, *n* is the predicted time step, and *i* is the current time step.

In Section 6, we describe the implementation and training steps of two networks; finally, we show the performance of the proposed DL models.

## 6. Experimental Results

### 6.1. Models Training and Optimization

The LSTM and BiLSTM models were trained on *production* and *consumption* datasets, respectively. Specifically, LSTM was trained to predict both the *DC Power* and *efficiency* of an inverter, computed adequately by the following equation:(6)$$\eta =\frac{P_{AC}}{P_{DC}}\times 100\%$$
where η is the efficiency of the inverter, expressed as a percentage, $P_{AC}$ is the AC power output of the inverter, measured in watts, and $P_{DC}$ is the DC power input of the inverter, measured in watts. AC power was retrieved to understand the actual energy production of the REC (Figure 8 shows the AC computed for a line of panels, each dot represents the value computed by the DL module from the data perceived). The BiLSTM was trained to predict GAP power by using all other consumption features.

For both models’ training, we used different and best hyperparameters. Table 1 presents the best hyperparameter settings used for the two DL networks. Each model had different settings, indicating that there is no one-size-fits-all solution when it comes to optimizing DL networks. Specifically, both networks were trained and optimized using the *mean squared error* (MSE) loss function and the *Adam* optimizer. We set the maximum number of epochs to 200 (with an *early stopping* technique set with a patience = 10) to prevent model overfitting, and the batch size to 72. We set the number of timesteps to 1 to capture short-term dependencies in the data. The learning rate α was set to 0.001, and the decay rate β was set to 0.00001 to ensure the effective optimization of the model.

For the LSTM model, we built a simpler architecture as a good starting point for our prediction tasks. Specifically, the better model architecture was based on one LSTM layer, two dense layers (one with 32 hidden units and a ReLU (*rectified linear unit*) activation function), and another dense layer with a single output unit and a sigmoid activation function. The BiLSTM with attention was implemented using a custom layer named *Attention*, which computes the attention weights and applies them to the input sequence. This layer takes the output of the previous layer (which is the output of the bidirectional LSTM layer) and computes a set of attention weights using a neural network with a single hidden layer. Specifically, the model consists of a bidirectional LSTM layer with 64 units, a dropout layer with a 0.2 dropout rate for regularization, and a batch normalization layer for training stability. Then, it includes an attention layer with 64 units to focus on relevant parts of the sequence, a dense layer with 32 units and ReLU activation, a dropout layer with a 0.3 dropout rate, and another batch normalization layer. Finally, there is a dense layer with 1 unit and sigmoid activation for predictions.

The learning curves depicted in Figure 9 show the DC and efficiency of the LSTM model over the course of training. As can be seen, as the number of epochs increases, the training and validation loss both trend towards zero, indicating that the LSTM model is able to capture the input–output relationship of the production dataset accurately.

#### Implementation Details

For the implementation of the two networks, we used Python program language version 3.8.12, Keras API version 2.4.3, to build and train our models, and Colab’s GPU to accelerate the training process. The data were preprocessed using Python libraries such as NumPy, Pandas, and Scikit-learn. The *MinMaxScaler* data normalization techniques were applied to the input features.

### 6.2. Testing on Consumption and Production Prediction

In our experiments, after the consumption dataset resampling to result in a 15 min steps GAP, the total number of samples was 138,352. The GAP behavior predicted with the BiLSTM-Attention model follows the real-time data, as shown in Figure 10.

Concerning energy production data elaboration, the LSTM acts on data related to DC and efficiency in reference to the lines of the panel. Figure 11 reports the data predicted by the LSTM for the DC and the efficiency η of each line. The graph analysis shows how the prediction follows the real behavior of the panel lines. The only significant anomaly is the absence of a second peak in the prediction when the real behavior presents two consecutive, very close peaks.

Table 2 reports the results obtained from LSTM (on production) to predict DC and η, and from BiLSTM-Attention (on consumption) to predict GAP features. The results show significant performances based on the RMSE and MAE metrics, suggesting that the two networks have a good predictive performance in predicting the associated features.

Table 3 reports an overview of which decision the TEANS module takes regarding the necessity of sending a request to reduce energy consumption to the REC members. The table reports some of the most meaningful time steps evaluated, with the estimated AC values, representing the sum of the three lines of panels and also the value of the whole REC GAP consumption computed (as explained in Section 5.1) from the hypothesis of 250 real estate units with the same energy requests. As it is clearly shown, the δ value is the discriminating factor in the decision; when δ is ≤0, a notification is delivered to the higher-consuming REC members (as described in functions (1) and (2)).

## 7. Conclusions

Our proposal enables better energy management and promotes efficient energy use. The solution discussed is meant to manage a renewable energy community by exploiting the energy produced without storing it. The proposed system can be applied to any such scenario without significant architectural modifications. Moreover, the reward system based on user behavior further incentivizes adopting sustainable energy practices, contributing to a more eco-friendly energy ecosystem. In conclusion, our research has successfully demonstrated the effectiveness of the proposed methodology in managing the energy produced among the community while minimizing reliance on the grid. This way, a REC aiming to minimize the fees due to energy obtained from the grid can adopt a complementary tool continuously monitoring the energy market for optimal buying and selling opportunities.

The current implementation of the orchestrator, a kind of complex event processing (CEP) system, has proven its value in sending notifications to users based on data collected and processed by AI models, guiding them to make informed decisions. Using AI in our research, primarily recurrent models, has been effective for short-term forecasting, yielding good energy consumption and production forecasting with the metrics employed, as shown and described in Section 6. Nevertheless, the system is strongly based on the assumption that users, or at least most of the REC’s members, cooperate and follow the notification received. Some other potential limitations of the proposed solution include the complexities of implementing the method in various RECs, computational requirements, and adaptability to changes in the energy landscape.

We envision several enhancements to our existing methodology for future work. First, we plan to integrate CEP capabilities into IoT devices, enabling more localized and automated decision making based on complex event patterns. Second, we aim to explore the potential of CEPs in automatically reducing power consumption in buildings by managing sub-meters as devices within a software-defined I/O framework, thus extending its capabilities beyond merely sending notifications.

Finally, we intend to investigate other deep learning techniques, such as convolutional neural networks or transformers, to improve anomaly detection in energy consumption patterns, complementing the DL models currently in use. By addressing these areas, we aim to build upon the foundational results laid out in this research, contributing to the ongoing development of sustainable and efficient energy management solutions. 

## Figures and Tables

**Figure 1 sensors-23-04549-f001:**
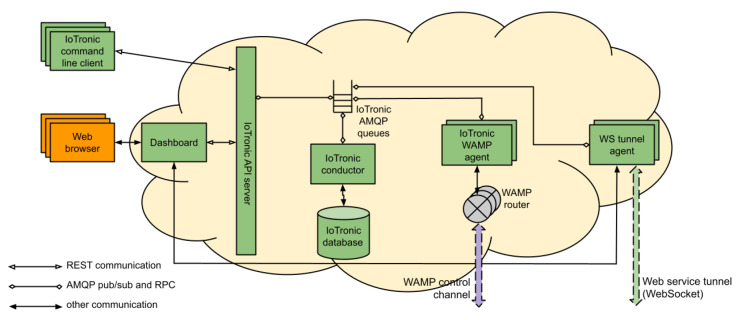
IoTronic’s architectural schema.

**Figure 2 sensors-23-04549-f002:**
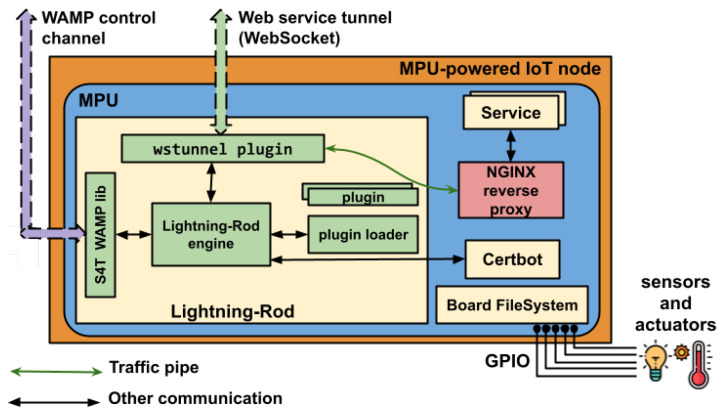
Lightning-Rod’s architectural schema.

**Figure 3 sensors-23-04549-f003:**
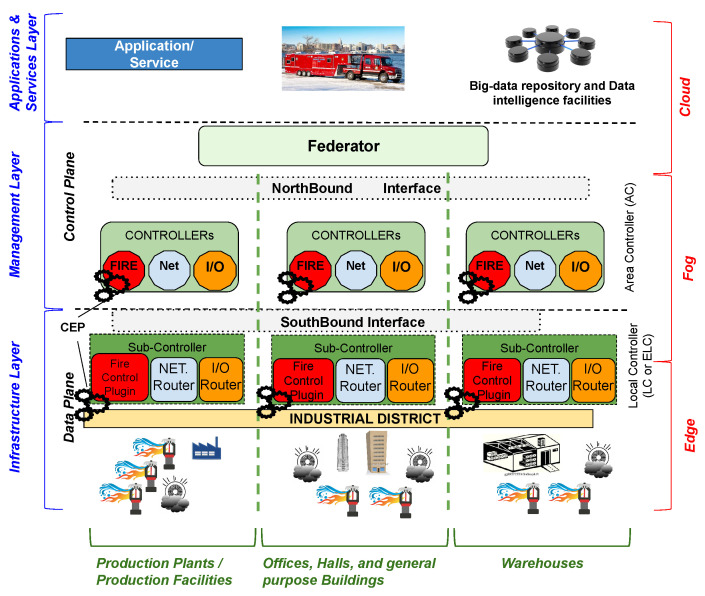
Example of a software-defined CPS acting as a self-conscious environment [50].

**Figure 4 sensors-23-04549-f004:**
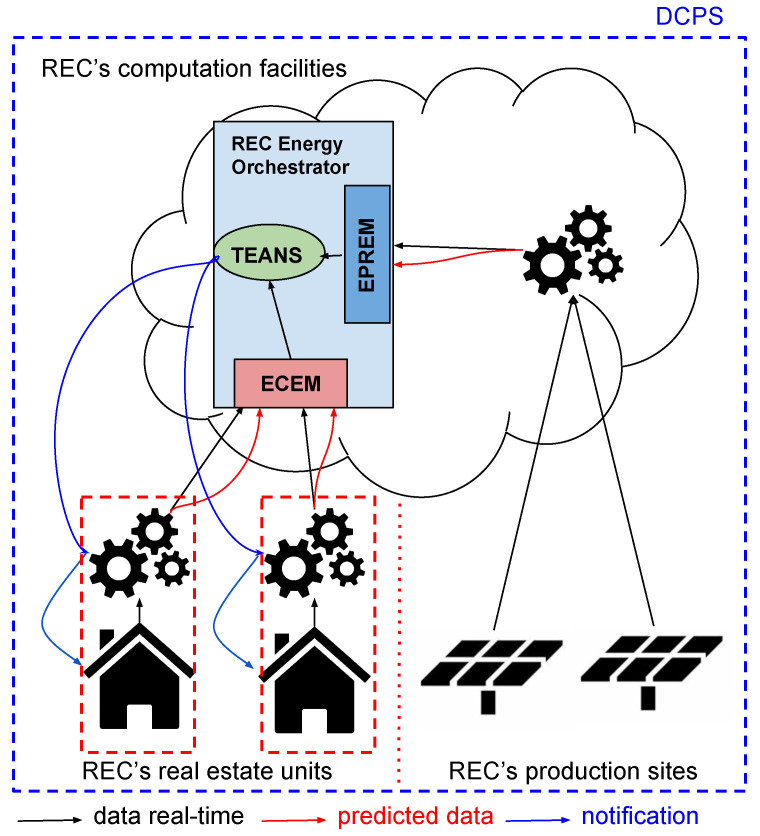
REC architecture. ECEM stands for Energy Consumption Estimation Module, EPREM stands for Energy Production Estimation Module, and TEANS stands for Threshold Evaluator and Notification System.

**Figure 5 sensors-23-04549-f005:**
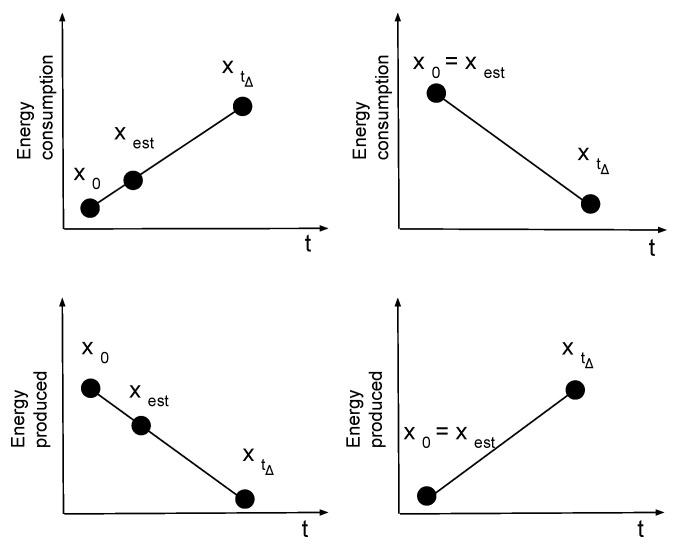
View of energy estimation value for consumption and production.

**Figure 6 sensors-23-04549-f006:**
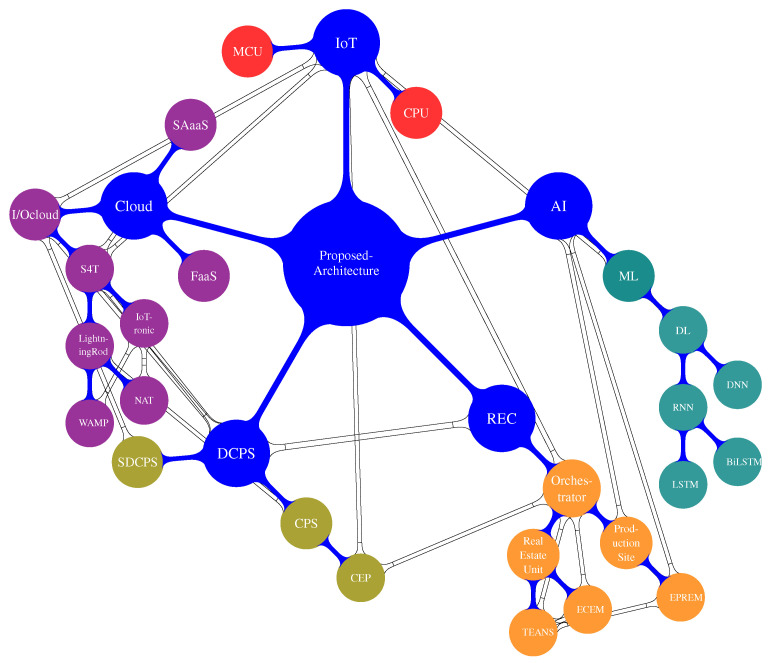
Technologies relationship chart.

**Figure 7 sensors-23-04549-f007:**
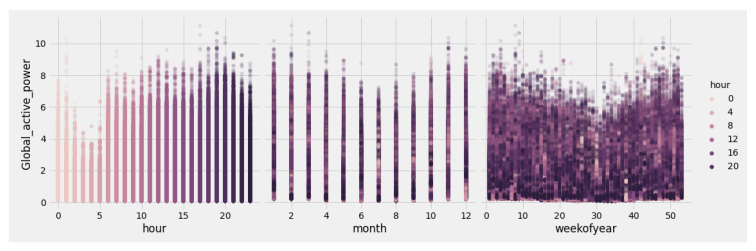
Gap seasonality of a real estate unit by hour, month, and week of the year.

**Figure 8 sensors-23-04549-f008:**
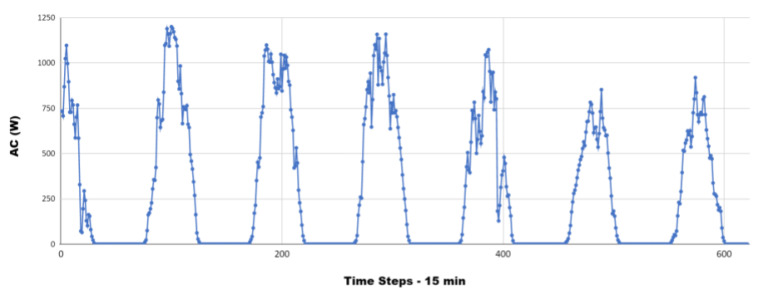
AC of a computed panel line.

**Figure 9 sensors-23-04549-f009:**
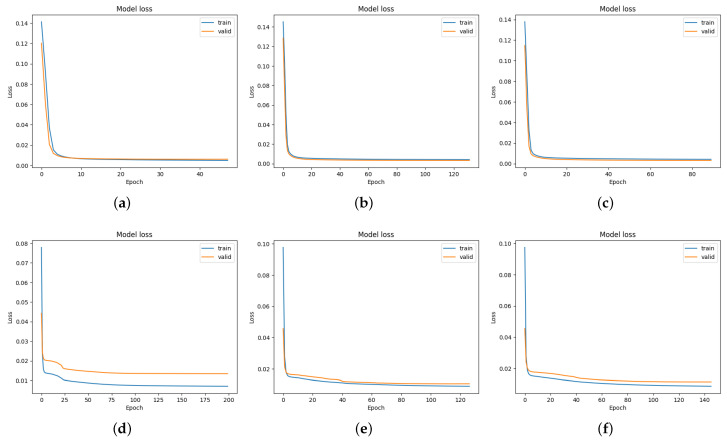
DC and efficiency training and validation loss curves of the LSTM model. (**a**) DC training and validation loss for the line of panels 1; (**b**) DC training and validation loss for the line of panels 2; (**c**) DC training and validation loss for the line of panels 3; (**d**) Efficiency training and validation loss for the line of panels 1; (**e**) Efficiency training and validation loss for the line of panels 2; (**f**) Efficiency training and validation loss for the line of panels 3.

**Figure 10 sensors-23-04549-f010:**
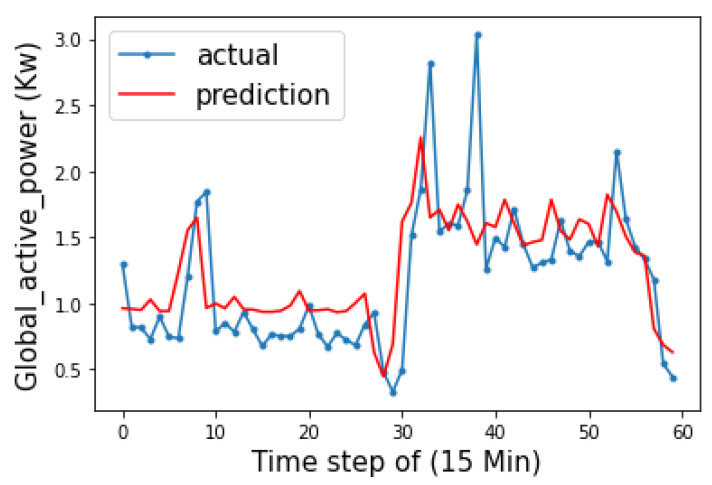
GAP predictions after dataset resampling of a real estate unit from 17 June at 17:00.

**Figure 11 sensors-23-04549-f011:**
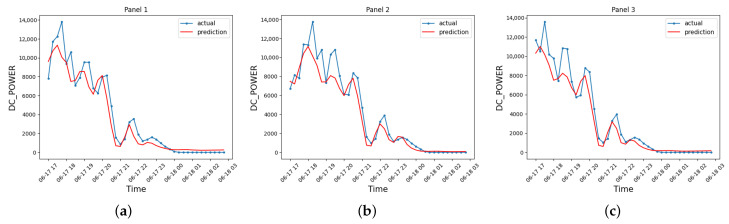
LSTM prediction graphs. (**a**) DC predicted by the model for the line of panels 1; (**b**) DC predicted by the model for the line of panels 2; (**c**) DC predicted by the model for the line of panels 3; (**d**) Efficiency predicted by the model for the line of panels 1; (**e**) Efficiency predicted by the model for the line of panels 2; (**f**) Efficiency predicted by the model for the line of panels 3.

**Table 1 sensors-23-04549-t001:** Hyperparameter settings.

Model	Hyperparameter	BestValue
LSTM	Learning rate	0.001
	Optimizer	“Adam”
	Decay rate	0.00001
	Loss function	“mean_squared_error”
	Number of layers	3 (LSTM, Dense, Dense)
	Units	[64, 32, 1]
	Activation function	“relu”,“sigmoid”
	Timesteps	1
	Maximum training epochs	200
	Early Stopping	Patience = 10, Monitor = loss
	Batch size	72
BiLSTM with Attention	Learning rate	0.001
	Optimizer	“Adam”
	Decay rate	0.00001
	Loss function	“mean_squared_error”
	Dropout	[0.2, 0.3]
	Number of layers	4 (Bidirectional LSTM, Attention, Dense, Dense)
	Units	[64, 64, 32, 1]
	Batch normalization	Present
	Activation function	“relu”,“sigmoid”
	Batch size	72

**Table 2 sensors-23-04549-t002:** Performance metrics on consumption (1EU) and production (3P).

Model	Feature	RMSE	MAE
*BiLSTM-Attention (consumption)*	GAP	0.457	0.299
LSTM(production(3P))	DC	909.19	509.53
LSTM(production(3P))	η	0.011	0.002

**Table 3 sensors-23-04549-t003:** Notification forwarded by TEANS according to δ computed.

Data	Time	AC Produced ${}^{1}$	GAP: All Real Estate Units ${}^{2}$	δ	Notification Sent
17 June	17:00	777.92	240.332816	**537.587184**	**No**
17 June	17:15	659.27	239.1358279	**420.1341721**	**No**
17 June	18:30	323.98	235.1232957	**88.85670433**	**No**
17 June	18:45	258.11	309.2678135	**−51.1578135**	**Yes**
17 June	19:00	36.8	411.623647	**−374.823647**	**Yes**
17 June	20:00	0.05	261.9870608	**−261.9370608**	**Yes**
17 June	21:00	0.01	233.4935617	**−233.4835617**	**Yes**
17 June	22:00	0	235.791142	**−235.791142**	**Yes**
17 June	23:00	0	234.9777611	**−234.9777611**	**Yes**

^1^ This value is the estimated value from the ECEM module; ^2^ This value is the estimated value from the EPREM module.

## Data Availability

Data supporting reported results can be found at https://archive.ics.uci.edu/ml/datasets/individual+household+electric+power+consumption (accessed on 10 January 2023) and https://www.kaggle.com/datasets/anikannal/solar-power-generation-data?select=Plant_1_Weather_Sensor_Data.csv (accessed on 10 January 2023).

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
