# Peer review of "A Deep Learning-Driven Self-Conscious Distributed Cyber-Physical System for Renewable Energy Communities"

_sensors, 2023, doi:10.3390/s23094549_

Round 1

Reviewer 1 Report

Attached

Minor comments. Please proof-read the manuscript.

Author Response

Dear Reviewer,

We would like to thank you for the time you spent reading our paper and providing valuable comments, which helped us to improve the quality of the paper. In the following, we provide details about how we dealt with your comments and summarize the modifications performed on the revised version of the paper. The changes with respect to the original version of the paper are highlighted in blue.

Following this careful revision, we believe that the paper has improved a lot. 

Reviewer #1, Concern #1: 

The paper would benefit from a clearer and more concise abstract that highlights the key contributions and findings of the research. 

Answer: 

We appreciate your suggestion to improve the clarity and conciseness of the abstract. Based on your feedback, we have revised the abstract to better emphasize the key contributions and findings of our research.

Reviewer #1, Concern #2: 

The paper should provide more details about the energy-aware edge-to-cloud architectural models and technologies, including how they were developed and how they work in practice.

Answer: 

We appreciate your suggestion to improve the architectural presentation. We add a sub-section (3.1.2) describing the IoTronic Cloud-side system that is used as the cornerstone of our solution. 

Reviewer #1, Concern #3:

The paper should explain more clearly how the proposed solution differs from existing solutions and what advantages it offers.

Answer: 

We appreciate the feedback and suggestion to clarify how the proposed solution differs from existing solutions and the advantages it offers. We have revised Section 2 (Related Works) to address this concern. In this section, we now provide a more comprehensive comparison between our approach and the existing literature, highlighting our proposed solution's unique contributions and advantages. These revisions will help better differentiate our solution from existing approaches and emphasize the value it brings to energy management and optimization.

Reviewer #1, Concern #4:

The paper should provide more details about the machine learning algorithms used in the system, including how they were developed and what data was used to train them.

Reviewer #1, Concern #6:

The paper should provide more details about the case study on optimizing renewable energy communities, including the data used and the evaluation metrics.

Answer: 

We greatly appreciate your input. As a result, we have restructured Section 5 to include a more comprehensive description of the case study, a thorough examination of the dataset, an explanation of the preprocessing methods, and a discussion of the deep learning models employed in the implementation of the system. Specifically, we created Sections 5.1, 5.1.1, and 5.2, to better organize the concepts of Section 6.

Reviewer #1, Concern #5:

The paper should provide more details about the microservices used in the system and how they were developed and deployed.

Answer: 

Thank you for your feedback. A deep analysis of the abstract has shown a misprint that refers to microservices that, in fact, are not considered at this point in our research. 

Reviewer #1, Concern #7:

The paper should provide more details about the experimental results, including statistical analysis and visualizations.

Answer: 

Thank you for your feedback. We agree with your suggestion that improves the overall quality of the discussion and enables a better understanding of our experiments. Specifically, we updated and extended Section 6, in particular, we also included a table (Table 1) containing the hyperparameters used in our experiments and a figure illustrating the learning curves. 

Reviewer #1, Concern #8:

The paper should discuss potential limitations and future directions for the proposed solution.

Answer: 

Thank you for your feedback. We have extended the discussion about potential limitations present in the Conclusion section, where a discussion about future work and direction is already present.

Reviewer #1, Concern #9: 

The paper should consider discussing the ethical implications of the proposed solution, including data privacy and security concerns.

Answer: 

Thank you for your suggestion. We modified section 4.1 according to your remarks a better explanation is provided, clarifying how the user's privacy and security are not affected by the Energy management workflows. 

Reviewer #1, Concern #10:

Some important papers are hereby provided to the authors to read and enrich their introduction/references: -

(i) Characterizing Current THD’s Dependency on Solar Irradiance and Supraharmonics Profiling for a Grid-Tied Photovoltaic Power Plant. Sustainability 2023, 15, 1214. https://doi.org/10.3390/su15021214

https://www.mdpi.com/2071-1050/15/2/1214 ….

(ii)      Grid-connected operation and control of single-phase asymmetrical multilevel inverter for distributed power generation; https://doi.org/10.1049/rpg2.12581 https://ietresearch.onlinelibrary.wiley.com/doi/full/10.1049/rpg2.12581

Answer: 

Thank you for your feedback. We have made the necessary changes to Section 2 of the related works according to your suggestions. In the revised text, we added the suggested papers. 

Reviewer 2 Report

This paper investigates deep learning based distributed cyber-physical system (DCPS) for renewable energy community. The authors claim that optimal implementation of an energy-aware DCPS is achieved and the transition toward a more sustainable society is supported. While the topic is interesting, the reviewer has the following comments/suggestions:

1. The description of the construction of RECs can be further improved. For example, list1 and list2 mentioned in Section 4.2 are not explained.
2. The pseudo-code readability of Algorithm 1 and Algorithm 2 needs to be improved, and the meanings of some variables are not explained.
3. In the experimental results section, the article focuses on load forecasting. The specific RECs can be established to illustrate the progressive nature of the proposed method.
4. Although there are 58 references, many of them are conference papers or online multimedia. Distributed energy system research from top journals may be discussed in the introduction, such as 10.1109/TSG.2022.3210014 and 10.1109/TSG.2014.2373150. But please note that this is not compulsory and please do judge by yourself.
5. There are many professional terms involved in this paper. It is suggested to explain their relationship using charts to facilitate understanding.

Author Response

Dear Reviewer,

We would like to thank you for the time you spent reading our paper and providing valuable comments, which helped us to improve the quality of the paper. In the following, we provide details about how we dealt with your comments and summarize the modifications performed on the revised version of the paper. The changes with respect to the original version of the paper are highlighted in blue. Following this careful revision, we believe that the paper has improved a lot.

 Reviewer #2, Concern #3:

In the experimental results section, the article focuses on load forecasting. The specific RECs can be established to illustrate the progressive nature of the proposed method.

Answer: 

Thank you for your feedback. We agree with your suggestion that improves the overall quality of the discussion and enables a better understanding of our experiments. Specifically, we updated and extended Section 6, in particular, we also included a table (Table 1) containing the hyperparameters used in our experiments and a figure illustrating the learning curves. 

Reviewer #2, Concern #4:

Although there are 58 references, many of them are conference papers or online multimedia. Distributed energy system research from top journals may be discussed in the introduction, such as 10.1109/TSG.2022.3210014 and 10.1109/TSG.2014.2373150. But please note that this is not compulsory and please do judge by yourself.

Answer: 

Thank you for your feedback. We have made the necessary changes to Section 2 of the related works according to your suggestions. In the revised text, we added the suggested papers. 

Reviewer #2, Concern #1: 

The description of the construction of RECs can be further improved. For example, list1 and list2 mentioned in Section 4.2 are not explained.

Reviewer #2, Concern #2: 

The pseudo-code readability of Algorithm 1 and Algorithm 2 needs to be improved, and the meanings of some variables are not explained.

Answer: 

Thank you for your suggestion. We modified Section 4.2 according to your remarks; in the text, a better explanation is provided, and some misprints are updated; the same was done on the two algorithms listed in the chapter. 

Reviewer #2, Concern #5: 

There are many professional terms involved in this paper. It is suggested to explain their relationship using charts to facilitate understanding.

Answer: 

Thank you for your suggestion. We enhance the abbreviation section with the technologies relationship chart, reported in Figure 11.

Round 2

Reviewer 1 Report

Authors have addressed my concerns.

Minor

Reviewer 2 Report

Thanks for the revision. The authors have adequately addressed all my concerns.